# Associations of body fat percentage with C-reactive protein levels in Canadian adults with and without osteoarthritis: Findings from the Canadian Longitudinal Study on Aging (CLSA)

Kendal A. Marriott[1], Paul W. Stratford[2], Chris P. Verschoor[2,3], Dawn M. E. Bowdish[2], Marina Mourtzakis[1], Jaclyn N. Chopp-Hurley[4], Emily G. Wiebenga[1], Monica R. Maly [1]*

1 University of Waterloo, Waterloo, Ontario, Canada, 2 McMaster University, Hamilton, Ontario, Canada, 3 Health Sciences North Research Institute, Sudbury, Ontario, Canada, 4 York University, Toronto, Ontario, Canada

* mrmaly@uwaterloo.ca

## Abstract

### Objectives

Determine the cross-sectional associations of serum inflammation with body composition and physical capacity in Canadian adults with/without osteoarthritis (OA) and multimorbidities.

### Methods

30,097 participants from the Canadian Longitudinal Study on Aging were categorized into eight subgroups (presence/absence of lower extremity OA, hand OA, multimorbidities), disaggregated by sex. Linear regression models (multiple covariates and log-transformed CRP) were completed with body composition and physical function measures as dependent variables. A priori, the research team defined that an increase in the adjusted $R^2$ between models by 1% or more was considered a meaningful change in explanatory power.

### Results

Higher CRP levels [standardized β(95%CI)] was associated with greater whole body fat percent [females 0.14 (0.13, 0.16), males 0.13 (0.12, 0.15)], trunk fat percent [females 0.16 (0.15, 0.18), males 0.13 (0.12, 0.15)] and lower appendicular lean mass index [males −0.12 (−0.13, −0.11)], independent of OA or multimorbidity.

### Conclusions

There were small but meaningful associations between CRP and each of percent adiposity and appendicular lean mass index. These associations were present regardless of OA and multimorbidity status. Percent adiposity, but not OA status, was

**Data availability statement:** Deidentified data are available from the Canadian Longitudinal Study on Aging (www.clsa-elcv.ca) for researchers who meet the criteria for access to de-identified CLSA data. Inquiries regarding data access and availability can be made via email (info@clsa-elcv.ca) and online (https://www.clsa-elcv.ca/data-access/).

**Funding:** The authors disclosed receipt of the following financial support for the research, authorship, and/or publication of this article: This research was made possible using the data/biospecimens collected by the Canadian Longitudinal Study of Aging (CLSA). Funding for the CLSA is provided by the Government of Canada, through the Canadian Institutes of Health Research (CIHR) under grant reference [LSA 94473] and the Canada Foundation for Innovation, as well as the following provinces, Newfoundland, Nova Scotia, Quebec, Ontario, Manitoba, Alberta, and British Columbia. This research has been conducted using the CLSA Baseline Comprehensive Dataset 5.1, under Application Number 2002013. The CLSA is led by Drs. Parminder Raina, Christina Wolfson and Susan Kirkland. This work was supported by the Arthritis Society Postdoctoral Fellowship Salary Award [award #210000000039] to [KAM]; Canada Research Chair in Aging and Immunity to [DMEB]; Canadian Institutes of Health Research (CIHR) Fellowship Award to [JC-H]; and The Arthritis Society Stars Mid-Career Development Award funded by the Canadian Institutes of Health Research-Institute of Musculoskeletal Health and Arthritis to [MRM]. The funders had no role in study design, data collection and analysis, decision to publish, or preparation of the manuscript. KAM was the recipient of a CLSA Data Access Trainee Fee Waiver.

**Competing interests:** The authors have declared that no competing interests exist.

consistently associated with systemic inflammation, suggesting that adiposity driven inflammation may contribute to OA related health outcomes.

## Introduction

Osteoarthritis (OA) is a leading cause of pain [1,2], commonly affecting the knees, hips and hands [1,2]. OA in the lower extremity joints produces significant difficulty with walking and stair negotiation [2,3]. OA in the hands is linked with poor grip strength and challenges the ability to engage in activities of daily living [4,5]. Ultimately, regardless of which joints are affected, OA is associated with reduced physical function [6] and physical activity [3], which challenges typical engagement in life.

Obesity, characterized by BMI > 30 kg/m$^2$, is a powerful predictor of incident and worsening OA [7]. There are multiple mechanisms by which obesity contributes to OA, including abnormal biomechanics [8,9], and altered body composition and systemic inflammation [10–12]. Biomechanical consequences of obesity include increased joint load [8,10,11] and altered knee and ankle biomechanics during walking [8]. There are multiple mechanisms implicated in obesity-related systemic inflammation [12,13]. The deposition of adipocytes produces inflammatory adipokines, such as increased leptin, interleukin-6 and tumor necrosis factor-alpha as well as decreased adiponectin, which are associated with inflammatory biomarkers. These changes contribute to chronic low-grade inflammation in obesity [13,14]. A chronic, low-grade systematic inflammation also results from activation of inflammatory pathways by abnormal mechanical loading [10,11]. Thus, in OA, there are at least two pathways to increase systemic inflammation: through altered biomechanics that activate inflammatory pathways [10,11] and through greater adiposity secreting pro-inflammatory cytokines [10–12]. Together, the systemic inflammation can be detected as higher levels of the acute phase protein C-reactive protein (CRP) [12,15–17] which is elevated in many chronic health conditions [18–25]. The level of circulating CRP is proportional to the intensity of the inflammatory process [26].

While a meta-analysis shows that CRP is higher in OA compared to healthy counterparts [18], there are multiple potential explanations for elevated systemic inflammation in those with OA. First, it is unclear if higher CRP is linked with adiposity, a feature of body composition, commonly associated with OA; or if it is associated with OA disease and the joint tissue damage itself. Second, it is unclear if the relationship between CRP levels with physical function depends on the joint (knee, hip, hand) affected by OA. It is possible that OA in large weight-bearing joints, such as the hip and knee, contribute more to systemic inflammation compared to small joints in the hand. Third, in OA, higher CRP concentrations are linked with symptomatic and structural disease changes, including (i) worse pain [18–20], (ii) greater disease incidence [20] and (iii) faster disease progression based on radiography [21], although results are somewhat inconsistent between studies [18–21,27]. Whether a similar association with higher CRP levels exists for measures of physical function, mobility and physical activity in OA are unknown. Fourth, those with OA are more likely to

have one or more comorbidities (i.e., multimorbidity) compared to those without OA [28]; however it is unclear whether higher CRP levels are associated with OA disease or with multimorbidity.

The purpose of this study was to determine the cross-sectional associations of serum inflammation (CRP) with body composition and physical capacity in Canadian adults with and without lower extremity OA, hand OA, and multimorbidities. Given the frequent co-occurrence of chronic conditions in aging, understanding whether inflammation is uniquely driven by OA or shared with other comorbidities is critical. We hypothesized that higher levels of CRP would be associated with (i) greater adiposity, (ii) reduced lean mass, and (iii) poorer physical capacity (slower walk time, chair rise time, standing balance and reduced grip strength). We also hypothesized that these associations would be stronger in subgroups with OA compared to subgroups without OA; in females compared to males; and in those with knee or hip OA compared to hand OA. Although this study is exploratory regarding interaction effects between OA status, multimorbidity status and sex, it was adequately powered to investigate the main effects. By exploring the strength of associations between CRP with adiposity and physical function in those with OA disease and multimorbidity, these findings will point toward the potential mechanisms by which systemic inflammation affects daily life in adults (45–85 years) with OA.

## Methods

### I. Canadian Longitudinal Study on Aging

The Canadian Longitudinal Study on Aging (CLSA) is a longitudinal study evaluating qualitative and quantitative outcomes in 11 centers across Canada [29] in more than 50,000 adults (45–85 years). The initial collection was in 2012–2015 with follow-up evaluations occurring every 3 years. Approximately 20,000 participants belong to the Tracking cohort and another 30,000 participants belong to the Comprehensive cohort. This analysis was completed using baseline data from the Comprehensive cohort of the CLSA. Eligibility for recruitment includes: (i) the ability to respond in English or French and (ii) residence within 25–50 kilometers of one of 11 centers (University of Victoria, University of British Columbia, Simon Fraser University, University of Calgary, University of Manitoba, McMaster University, University of Ottawa, Research Institute of the McGill University Health Centre, Université de Sherbrooke, Dalhousie University, Memorial University of Newfoundland). CLSA participants from the Comprehensive cohort were recruited through several avenues, including provincial healthcare registration databases, random digit dialing and the Quebec Longitudinal Study on Nutrition and Successful Aging (NuAge). Data for the Comprehensive cohort was collected from May 2012 to May 2015 through face-to-face interviews at home and data collection sites [29]. A subset of these data were made available to this research team after application and approval to the CLSA Data and Sample Access Committee (DSAC). Further, this research team accessed information of how data were collected from CLSA documentation. This article contains a summary of the CLSA methods and identifies the associated standard operating procedures documented by CLSA for reference. Ethics approval for data analysis was obtained from the University of Waterloo Research Ethics Committee (#42356). Data were accessed October 30 2020. Authors did not have access to information that could identify participants during or after collection.

### II. Participants

Data from all participants in the Comprehensive cohort of the CLSA were eligible for inclusion. However, for each of the separate linear regression models, only data from participants with completed measurements of interest were included in the analyses (63–67% depending on the variable of interest); those with partial data were excluded.

All participants were categorized into subgroups according to the (i) total lower extremity OA score, (ii) total hand OA score, and (iii) total multimorbidity score. The *total lower extremity OA* score was calculated based on two questions that asked participants to self-report a diagnosis of knee OA and hip OA. Participants were categorized as having lower extremity OA if they self-reported a diagnosis of (i) knee OA only, (ii) hip OA only or (iii) both knee and hip OA. Therefore, if a participant did not self-report a diagnosis of knee OA or hip OA, this participant was classified as having "no lower

extremity OA" ([Fig 1]). The *total hand OA* score was calculated based on one question that asked participants to self-report a diagnosis of hand OA. Therefore, if a participant did not self-report a diagnosis of hand OA, this participant was classified as having "no hand OA" ([Fig 1]).

A *total multimorbidity* score was calculated based on self-reported diagnosis of 21 multimorbidities: rheumatoid arthritis; Multiple Sclerosis; Alzheimer's disease or dementia; Parkinson's disease; emphysema, chronic bronchitis, chronic obstructive pulmonary disorder or chronic lung changes due to smoking; ministroke or transient ischemic attack; stroke or cerebrovascular accident; effects of stroke, cerebrovascular accident or transient ischemic attack; hypertension; heart disease including chronic heart failure; angina; heart attack or myocardial infarction; shortness of breath following strenuous activity; diabetes or borderline diabetes; peripheral vascular disease; epilepsy; mood disorders; clinical depression; intestinal/stomach ulcers; cancer; bowel disorders. Participants were categorized as having multimorbidity if they self-reported a diagnosis of 2 or more of the listed 21 conditions [30]. A categorization of the presence or absence was selected based on previous evidence demonstrating an association between multimorbidity and worsening pain and performance-based physical function in individuals with knee and/or hip OA [3,6,28,31]. Therefore, if a participant did not self-report a diagnosis of 2 or more of the listed conditions, this participant was classified as having "no multimorbidity" ([Fig 1]). Based on the total lower extremity OA score, total hand OA score and total multimorbidity score, participants were categorized into eight subgroups ([Fig 1]). Participants in subgroup A self-reported none of these conditions and thus, subgroup A is a healthy reference group in this sample. In contrast, participants in subgroup H self-reported the presence of all three of these health conditions. The subgroups in between A and H self-reported various combinations of lower extremity OA, hand OA and multimorbidity. For example, individuals with both lower extremity OA and hand OA but differing multimorbidity status, are represented by subgroups G and H.

## III. Independent measures

**Measurement of C-reactive protein (CRP).** CRP (mg/L) was obtained at a CLSA data collection site through collection of venous blood from the antecubital fossa according to previously published protocols (SOP_BCP_0001, SOP_BCP_0003, [S1 File]). Circulating CRP protein was measured in serum using a standard immunoturbidimetric assay (Roche Cobas 8000 analyzer), quantitated using a high-sensitivity ELISA and used as a measure of systemic inflammation.

## IV. Dependent measures

**Grip strength.** Grip strength (kg) was measured using a Tracker Freedom Wireless Grip Dynamometer (SOP_DCS_0028, [S1 File]). The average strength is more reliable than the peak strength [32] and therefore, the average strength between three repeated trials was selected for analysis.

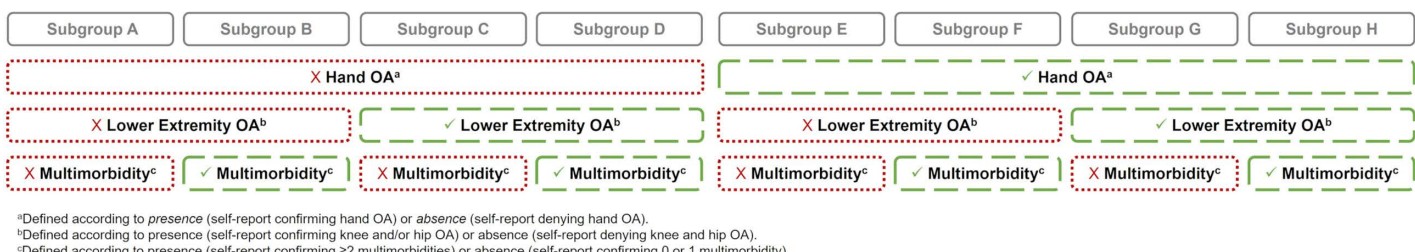

[a]Defined according to *presence* (self-report confirming hand OA) or *absence* (self-report denying hand OA).
[b]Defined according to presence (self-report confirming knee and/or hip OA) or absence (self-report denying knee and hip OA).
[c]Defined according to presence (self-report confirming ≥2 multimorbidities) or absence (self-report confirming 0 or 1 multimorbidity).

**Fig 1. Subgroup categorization based on total hand OA score, total lower extremity OA score and total multimorbidity score.** Multimorbidity score represents those with 2 or more self-reported diagnoses across 21 health conditions. Green (— —) indicates the subgroup has the health condition. Red (– – –) indicates the subgroup does not have the health condition. Subgroups range from healthiest (Subgroup A represents those without OA or multimorbidity) to the least healthy (Subgroup H represents those with lower extremity OA, hand OA and multimorbidity).

**Mobility index.** A mobility index (unitless) was created using an aggregate score from four variables: (i) *4-meter walk* (total time to walk a distance of 4 meters), (ii) *one-leg standing balance* (best time attained for standing on one leg, up to a maximum 60 seconds), (iii) *chair rise* (average time to rise out of a chair 5 times), and (iv) *timed-up-and-go* (time to rise out of a chair, walk 3 meters and return to a seated position in the chair) (S1 File: SOP_DCS_0021, SOP_DCS_0024, SOP_DCS_0023, SOP_DCS_0022, respectively). Each of the four variables was measured in seconds and converted to standard normal variables using Equation 1. The sum of the four variables was then calculated while accounting for the direction of each variable (Equation 2) [33].

$$norm\_var_{x,i} = \frac{var_{x,i} - \mu_{Var_x}}{\sigma_{var_x}} \tag{1}$$

Where,

$norm\_var_{x,i}$ is the normalized value of a particular variable, $x$, for participant, $i$ ($i=1$: number of participants)

$var_{x,i}$ is the value of that particular variable, $x$, for participant, $i$

$\mu_{Var_x}$ is the mean of that particular variable, $x$

$\sigma_{var_x}$ is the standard deviation of that particular variable, $x$

$$Mobility\ Index = -[norm_{wlk} + norm_{cr} + norm_{timed-up-and-go} + (-norm_{bal})] \tag{2}$$

Where,

$norm_{wlk}$ is the normalized walk time

$norm_{cr}$ is the normalized chair rise time

$norm_{timed-up-and-go}$ is the normalized tug time

$norm_{bal}$ is the normalized standing balance time

**Fat mass, fat percent, lean mass.** The following variables were acquired: Dual-Energy X-Ray Absorptiometry (DXA) whole-body fat mass (g) (calculated using fat mass and weight), whole-body fat percent (%), whole-body lean mass (g), percent of fat tissue to total tissue mass in trunk (%), appendicular fat mass index (kg/m$^2$) and appendicular lean mass index (kg/m$^2$). Participant height (m) and weight (kg) were obtained before proceeding with collection of whole-body and regional body composition measured using dual-energy x-ray absorptiometry (DXA) (Hologic Discovery QDR 4500) at a CLSA data collection site according to previously published protocols (SOP_DCS_0017, S1 File). Participants were assisted onto the table and positioned supine, with their spine straight and their entire body within the scanning limits. The legs were positioned together with the feet plantarflexed, separated and toes pointed inwards. A strap was used to secure the legs in this position. Arms were positioned straight alongside the body, while maintaining separation from the body; forearms pronated (palms facing down) and elbows slightly flexed. Participants were instructed to close their eyes and remain still throughout the exam.

## V. Covariates

Seven covariates were included in the analyses. These covariates were selected based on their ability to explain variance in body composition, physical capacity and general health in individuals with OA. Covariates included age (yrs) [34], body mass index (BMI) (kg/m$^2$) [35], education level (categorical) [31], income level (categorical) [36], social inequality (1–10) [1], scores on the Center for Epidemiological Studies Short Depression scale (CES-D 10) (0–30) [37] and subgroup which was defined according to the presence of OA in the lower extremity, OA in the hands and the number of multimorbid conditions (categorical, Fig 1). While BMI is related to CRP [15], BMI was included as a covariate to enable the interpretation of findings to be attributed to the role of circulating inflammation rather than overall body size. Education level was classified into seven categories, with a score of 1 indicating no post-secondary degree/certificate/diploma, a score of 6

indicating university degree or certificate above bachelor's degree and a score of 7 indicating *other*. Income level was classified into five brackets, with a score of 1 indicating an annual household income less than $20,000 and a score of 5 indicating an annual household income greater than $150,000. Social inequality was evaluated through self-report of perceived level of social standing on a 10-point scale, with a score of 1 indicating lowest standing in the community and a score of 10 indicating highest standing in the community.

### VI. Data and statistical analysis

Sex differences exist in OA prevalence [1] and symptoms [34]. Therefore, analyses were completed separately for females and males. There were a total of 8 subgroups (A through H), with the 'healthiest' participants in subgroup A (no lower extremity OA, no hand OA, no multimorbidities) and the 'least healthy' participants in subgroup H (lower extremity OA, hand OA and multimorbidities). The independent variable (CRP) was log-transformed because of a skewed distribution. There were eight dependent variables, each explored in independent analyses: grip strength, mobility index, whole body fat mass, whole body lean mass, whole body fat percent, trunk fat percent, appendicular fat mass index, appendicular lean mass index. For each of these eight dependent measures, three linear regression models were completed to determine the association between the independent variable (log-CRP) and the dependent variable. For each dependent measure, the first model was comprised exclusively of seven covariates (age, BMI, education level, income level, social inequality, CES-D, subgroup). The second model included log-CRP, which was entered following the terms from model 1. The third model included an interaction term (log-CRP × subgroup) which was entered following the terms from model 1 and model 2. A log-CRP × subgroup interaction term was considered to determine whether the association of CRP with each dependent variable was influenced by the presence or absence of OA and multimorbidity. Subgroup was considered a factored variable with subgroup A designated as the reference category. Therefore, for the third model, subgroups B through H were compared to subgroup A. Adjusted $R^2$ values were calculated to determine whether separate addition of the independent variable (log-CRP) and interaction term (log-CRP × subgroup) explained variance in the dependent variable over and above the first model (covariates only) and second model (covariates + log-CRP), respectively.

Due to the large sample size (and the increased likelihood of rejecting the null hypothesis when the null hypothesis is true), the adjusted $R^2$ value had to increase by 1% or more in order for the predictor to be considered a meaningful change in explanatory power. An $R^2$ value of 1% represents the threshold for a small effect [38]. A Benjamini-Hochberg correction for multiple comparisons was applied. Specifically, correction was applied to the *p*-values for the CRP term obtained from the overall analysis of variance (ANOVA) table associated with each linear regression model as well as the *p*-values for the β-coefficients associated with the CRP term. The corrections were applied across the eight dependent variables (thus accounting for the multiple comparisons) but clustered within sex (female, male) and model (model 2, model 3). Overall, an adjusted $R^2$ increase ≥1% were used as thresholds for practical and meaningful change, in addition to p-values corrected for multiple comparisons. Several model diagnostic tests were performed and plots (residuals versus fitted, normal Q-Q, scale-location, residuals versus leverage) were visually inspected to confirm normality, heteroscedasticity and absence of collinearity. Analyses were conducted using complete case analysis; no imputation was performed. All statistical analyses were completed using R (Version 4.12).

## Results

### I. Participants

Data from 30,097 participants were categorized into 1 of 8 subgroups (Fig 1). The percentage of females with lower extremity OA, hand OA and multimorbidity was 22.4%, 16.6% and 46.4%, respectively. The percentage of males with lower extremity OA, hand OA and multimorbidity was 15.7%, 7.7% and 45.0%, respectively. For each dependent variable, the subset of participants that had complete data sets were included in the statistical analyses. The flowchart illustrating the inclusion and exclusion of participants is presented in Fig 2. The demographic characteristics of the included participants

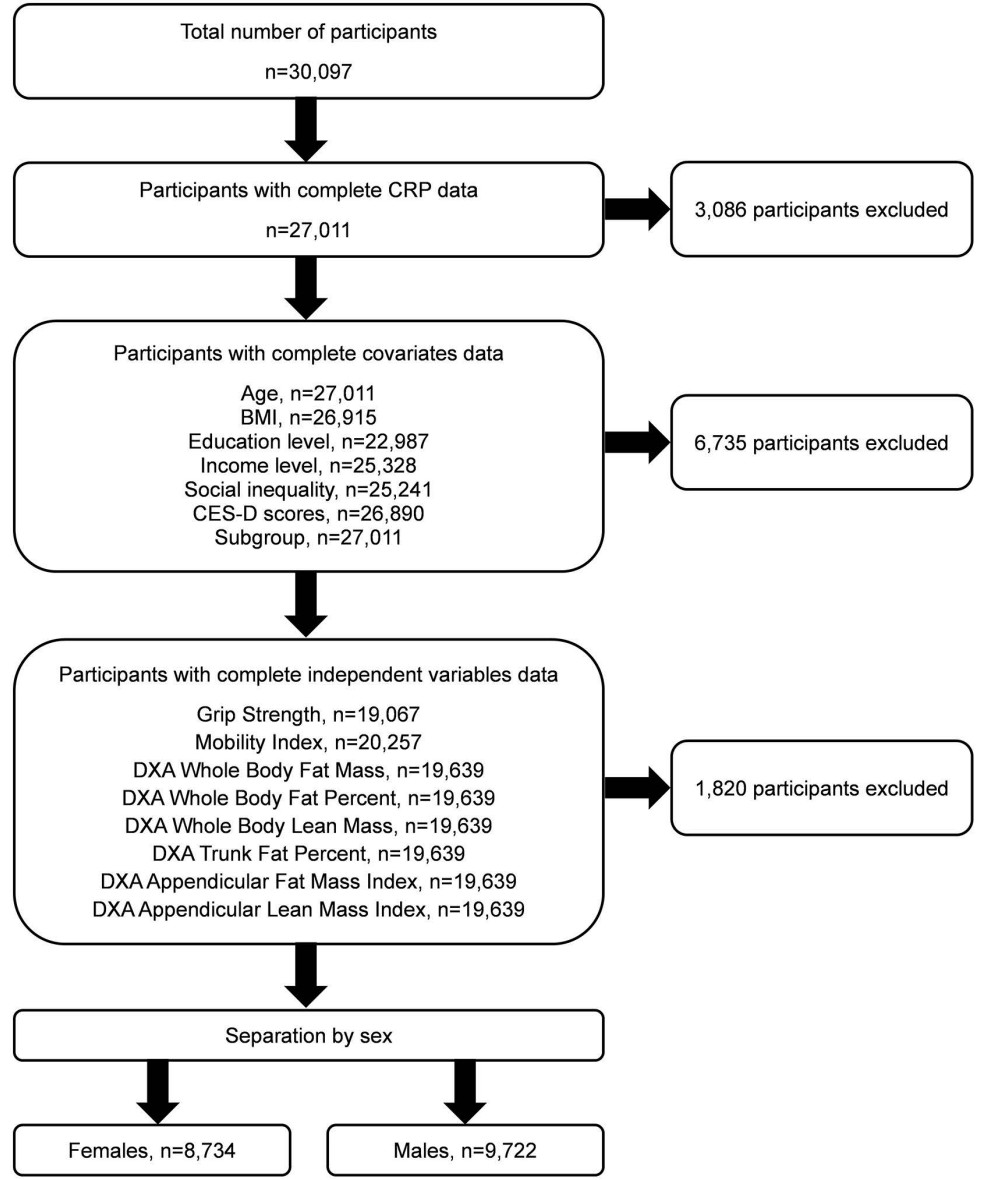

**Fig 2. Flowchart of participant inclusion and exclusion, based on availability of complete data.** *Subgroup* refers to self-report status (yes/no) for the presence of lower extremity osteoarthritis (OA), hand OA and multimorbidity. *CRP*, C-reactive protein; *CES-D*, Center for Epidemiological Studies Short Depression scale; *DXA*, Dual-Energy X-Ray Absorptiometry.

for each dependent variable is presented in S2 File. The number of participants in each subgroup for each dependent variable is presented in S3 File. The distribution of CRP values (mg/L) in females and males is presented in Fig 3. The mean and standard deviation (SD) of CRP values for each subgroup (A to H) for females and males are presented in Table 1.

## II. Model outcomes

Model 1 (covariates only) explained a meaningful amount of the variance for all dependent variables, in both females and males.

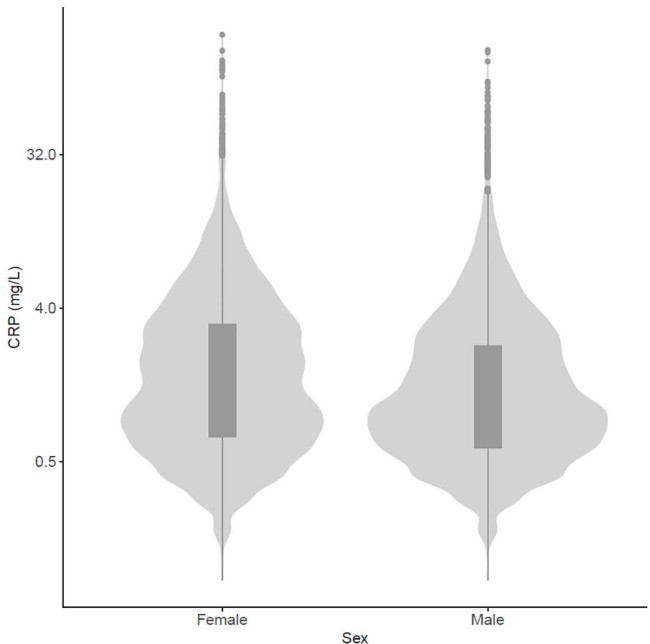

**Fig 3. Violin plot showing the distribution of CRP values (mg/L) (vertical axis), separated by sex (horizontal axis).** A total of 541 (4.0%) females and 386 (2.9%) males had levels greater than 10 mg/L, which in a clinical context, are indicative of clinically significant inflammation.

**Table 1. Mean and standard deviation (SD) of C-Reactive Protein (CRP) values for each subgroup (A to H) for females and males.**

| Subgroup | Females Mean (SD) | Males Mean (SD) |
|---|---|---|
| A | 2.24 (4.54) | 1.96 (4.39) |
| B | 3.14 (5.73) | 2.57 (4.99) |
| C | 2.72 (4.79) | 1.95 (3.78) |
| D | 3.70 (5.60) | 3.06 (5.43) |
| E | 2.15 (5.21) | 2.24 (5.12) |
| F | 2.82 (3.85) | 2.55 (5.14) |
| G | 2.56 (3.45) | 2.44 (4.55) |
| H | 4.01 (7.95) | 4.12 (8.34) |

In model 2, for 5 of the 8 dependent variables (grip strength, mobility index, whole body fat mass, whole body lean mass, appendicular fat mass index), no meaningful change in the explained variance of the model (represented by an increase in the adjusted $R^2$ by 1% or more) was observed with the addition of log-CRP (Table 2). Detailed results for the remaining 3 dependent variables (DXA whole body fat percent, DXA trunk fat percent, DXA appendicular lean mass index) are presented below. The unstandardized and standardized β coefficients for log-CRP for model 2 (where log-CRP was added to the regression) are presented in Table 3 and for model 3 (where log-CRP × subgroup interaction term was added to the regression) are presented in Table 4. An example of the unstandardized and standardized β coefficients for covariates is presented in Table 5. Additionally, for females and males, a plot visually comparing the standardized β coefficients for log-CRP for model 2 for all outcomes, are presented in Fig 4. The magnitudes of change in the explained variance

Table 2. Regression models for females and males used to explain grip strength, mobility index, DXA whole body fat mass, DXA whole body fat percent, DXA whole body lean mass, DXA trunk fat percent, DXA appendicular fat mass index, DXA appendicular lean mass index.

| Females | Model 1 (covariates† only) | | Model 2 (covariates†+log-CRP) | | Model 3 (covariates†+log-CRP+log-CRP×sub-group) | |
|---|---|---|---|---|---|---|
| | Explained Variance | Model Significance | Explained Variance | Model Significance | Explained Variance | Model Significance |
| Grip Strength (kg) | $R^2$=0.225 Adj $R^2$=0.224 | $F_{(21, 8985)}$ = 124.54 p<0.01‡ | $R^2$=0.231 Adj $R^2$=0.229 | $F_{(22, 8984)}$ = 122.86 p<0.01‡ | $R^2$=0.232 Adj $R^2$=0.230 | $F_{(29, 8977)}$ = 93.76 p<0.01‡ |
| Mobility Index | $R^2$=0.341 Adj $R^2$=0.339 | $F_{(21, 9743)}$ = 239.57 p<0.01‡ | $R^2$=0.343 Adj $R^2$=0.341 | $F_{(22, 9742)}$ = 230.73 p<0.01‡ | $R^2$=0.344 Adj $R^2$=0.342 | $F_{(29, 9735)}$ = 175.77 p<0.01‡ |
| DXA Whole Body Fat Mass (g) | $R^2$=0.878 Adj $R^2$=0.877 | $F_{(21, 9455)}$ = 3229.39 p<0.01‡ | $R^2$=0.879 Adj $R^2$=0.879 | $F_{(22, 9454)}$ = 3133.09 p<0.01‡ | $R^2$=0.879 Adj $R^2$=0.879 | $F_{(29, 9447)}$ = 2383.63 p<0.01‡ |
| DXA Whole Body Fat Percent (%) | $R^2$=0.599 Adj $R^2$=0.598 | $F_{(21, 9455)}$ = 673.48 p<0.01‡ | **R2=0.614** **Adj R2=0.613***  | $F_{(22, 9454)}$ = 683.90 p<0.01‡ | $R^2$=0.617 Adj $R^2$=0.616 | $F_{(29, 9447)}$ = 525.79 p<0.01‡ |
| DXA Whole Body Lean Mass (g) | $R^2$=0.600 Adj $R^2$=0.599 | $F_{(21, 9455)}$ = 674.85 p<0.01‡ | $R^2$=0.605 Adj $R^2$=0.604 | $F_{(22, 9454)}$ = 657.53 p<0.01‡ | $R^2$=0.605 Adj $R^2$=0.604 | $F_{(29, 9447)}$ = 499.18 p<0.01‡ |
| DXA Trunk Fat Percent (%) | $R^2$=0.605 Adj $R^2$=604 | $F_{(21, 9455)}$ = 688.68 p<0.01‡ | **R2=0.624** **Adj R2=0.623***  | $F_{(22, 9454)}$ = 713.18 p<0.01‡ | $R^2$=0.627 Adj $R^2$=0.626 | $F_{(29, 9447)}$ = 548.63 p<0.01‡ |
| DXA Appendicular Fat Mass Index (kg/m²) | $R^2$=0.800 Adj $R^2$=0.800 | $F_{(21, 9455)}$ = 1806.19 p<0.01‡ | $R^2$=0.801 Adj $R^2$=0.800 | $F_{(22, 9454)}$ = 1762.12 p<0.01‡ | $R^2$=0.801 Adj $R^2$=0.801 | $F_{(29, 9447)}$ = 1313.61 p<0.01‡ |
| DXA Appendicular Lean Mass Index (kg/m²) | $R^2$=0.700 Adj $R^2$=0.700 | $F_{(21, 9455)}$ = 1049.37 p<0.01‡ | $R^2$=0.708 Adj $R^2$=0.707 | $F_{(22, 9454)}$ = 1041.86 p<0.01‡ | $R^2$=0.708 Adj $R^2$=0.707 | $F_{(29, 9447)}$ = 791.10 p<0.01‡ |

| Males | Explained Variance | Model Significance | Explained Variance | Model Significance | Explained Variance | Model Significance |
|---|---|---|---|---|---|---|
| Grip Strength (kg) | $R^2$=0.229 Adj $R^2$=0.227 | $F_{(21, 10037)}$ = 141.86 p<0.01‡ | $R^2$=0.234 Adj $R^2$=0.232 | $F_{(22, 10036)}$ = 139.47 p<0.01‡ | $R^2$=0.235 Adj $R^2$=0.232 | $F_{(29, 10029)}$ = 106.03 p<0.01‡ |
| Mobility Index | $R^2$=0.256 Adj $R^2$=0.254 | $F_{(21, 10469)}$ = 171.32 p<0.01‡ | $R^2$=0.258 Adj $R^2$=0.256 | $F_{(22, 10468)}$ = 165.35 p<0.01‡ | $R^2$=0.260 Adj $R^2$=0.257 | $F_{(29, 10461)}$ = 126.42 p<0.01‡ |
| DXA Whole Body Fat Mass (g) | $R^2$=0.801 Adj $R^2$=0.800 | $F_{(21, 10139)}$ = 1941.93 p<0.01‡ | $R^2$=0.805 Adj $R^2$=0.805 | $F_{(22, 10138)}$ = 1902.39 p<0.01‡ | $R^2$=0.805 Adj $R^2$=0.805 | $F_{(29, 10131)}$ = 1443.86 p<0.01‡ |
| DXA Whole Body Fat Percent (%) | $R^2$=0.550 Adj $R^2$=0.549 | $F_{(21, 10139)}$ = 590.95 p<0.01‡ | **R2=0.565** **Adj R2=0.565***  | $F_{(22, 10138)}$ = 599.72 p<0.01‡ | $R^2$=0.566 Adj $R^2$=0.565 | $F_{(29, 10131)}$ = 455.79 p<0.01‡ |
| DXA Whole Body Lean Mass (g) | $R^2$=0.565 Adj $R^2$=0.564 | $F_{(21, 10139)}$ = 626.54 p<0.01‡ | $R^2$=0.569 Adj $R^2$=0.568 | $F_{(22, 10138)}$ = 607.89 p<0.01‡ | $R^2$=0.569 Adj $R^2$=0.568 | $F_{(29, 10131)}$ = 461.21 p<0.01‡ |
| DXA Trunk Fat Percent (%) | $R^2$=0.563 Adj $R^2$=0.562 | $F_{(21, 10139)}$ = 621.53 p<0.01‡ | **R2=0.579** **Adj R2=0.578***  | $F_{(22, 10138)}$ = 632.54 p<0.01‡ | $R^2$=0.580 Adj $R^2$=0.579 | $F_{(29, 10131)}$ = 482.70 p<0.01‡ |
| DXA Appendicular Fat Mass Index (kg/m²) | $R^2$=0.735 Adj $R^2$=0.734 | $F_{(21, 10139)}$ = 1338.82 p<0.01‡ | $R^2$=0.737 Adj $R^2$=0.736 | $F_{(22, 10138)}$ = 1291.06 p<0.01‡ | $R^2$=0.737 Adj $R^2$=0.737 | $F_{(29, 10131)}$ = 981.05 p<0.01‡ |
| DXA Appendicular Lean Mass Index (kg/m²) | $R^2$=0.656 Adj $R^2$=0.656 | $F_{(21, 10139)}$ = 922.49 p<0.01‡ | **R2=0.669** **Adj R2=0.668***  | $F_{(22, 10138)}$ = 931.57 p<0.01‡ | $R^2$=0.669 Adj $R^2$=0.668 | $F_{(29, 10131)}$ = 707.49 p<0.01‡ |

Note. *CRP*, C-reactive protein; *DXA*, Dual-Energy X-Ray Absorptiometry.

**Bolded*** Indicates a meaningful increase in explained variance with addition of the predictor to the model (adjusted $R^2$ value increased by 1% or more).

†Covariates included *age, body mass index, education level, income level, social inequality, scores on the CES-D and subgroup.*

‡Indicates a Benjamini-Hochberg correction for multiple comparisons was applied.

**Table 3. Unstandardized and standardized β coefficients (95%CI) for log-CRP for model 2 (covariates†+log-CRP) for females and males.**

| | Model 2 *Unstandardized* log-CRP β Coefficient (95%CI) | Model 2 *Standardized* log-CRP β Coefficient (95%CI) |
|---|---|---|
| Females | | |
| Grip Strength (kg) | −0.52 (−0.65, −0.40)‡ | −0.09 (−0.11, −0.07)‡ |
| Mobility Index | −0.14 (−0.19, −0.09)‡ | −0.05 (−0.07, −0.03)‡ |
| DXA Whole Body Fat Mass (g) | 506.48 (421.59, 591.37)‡ | 0.05 (0.04, 0.06)‡ |
| DXA Whole Body Fat Percent (%) | 0.89 (0.80, 0.98)‡ | 0.14 (0.13, 0.16)‡ |
| DXA Whole Body Lean Mass (g) | −575.55 (−679.34, −471.76)‡ | −0.08 (−0.10, −0.07)‡ |
| DXA Trunk Fat Percent (%) | 1.23 (1.12, 1.34)‡ | 0.16 (0.15, 0.18)‡ |
| DXA Appendicular Fat Mass Index (kg/m$^2$) | 0.03 (0.01, 0.05)‡ | 0.02 (0.01, 0.03)‡ |
| DXA Appendicular Lean Mass Index (kg/m$^2$) | −0.11 (−0.13, −0.10)‡ | −0.11 (−0.12, −0.09)‡ |
| Males | | |
| Grip Strength (kg) | −0.80 (−0.99, −0.61)‡ | −0.08 (−0.10, −0.06)‡ |
| Mobility Index | −0.14 (−0.19, −0.09)‡ | −0.05 (−0.07, −0.03)‡ |
| DXA Whole Body Fat Mass (g) | 643.91 (557.68, 730.13)‡ | 0.07 (0.06, 0.08)‡ |
| DXA Whole Body Fat Percent (%) | 0.77 (0.69, 0.85)‡ | 0.13 (0.12, 0.15)‡ |
| DXA Whole Body Lean Mass (g) | −641.47 (−770.66, −512.28)‡ | −0.07 (−0.08, −0.05)‡ |
| DXA Trunk Fat Percent (%) | 0.93 (0.83, 1.02)‡ | 0.13 (0.12, 0.15)‡ |
| DXA Appendicular Fat Mass Index (kg/m$^2$) | 0.06 (0.04, 0.07)‡ | 0.05 (0.04, 0.06)‡ |
| DXA Appendicular Lean Mass Index (kg/m$^2$) | −0.14 (−0.16, −0.13)‡ | −0.12 (−0.13, −0.11)‡ |

Note. *CRP*, C-reactive protein; *DXA*, Dual-Energy X-Ray Absorptiometry.

†Covariates included *age*, *body mass index*, *education level*, *income level*, *social inequality*, *scores on the CES-D* and *subgroup*.

‡Indicates a Benjamini-Hochberg correction for multiple comparisons was applied.

were small for 5 of the 8 dependent variables (indicated by an increase in the adjusted $R^2$ by less than 1%) and therefore, the β coefficients for these dependent variables were not considered important.

In model 3, for each of the dependent variables, no meaningful change in the explained variance (represented by an increase in the adjusted $R^2$ by 1% or more) was observed with the addition of the interaction term (log-CRP×subgroup) to the regression (Table 2, Table 4).

**DXA whole body fat percent.** For females, model 1 (covariates only) explained 59.8% of the variance in DXA whole body fat percent (Table 1). When log-CRP was added to model 1, there was a meaningful increase in the explained variance of the model (model 2) (Table 1). Specifically, after adjusting for multiple comparisons, a greater log-CRP level was associated with greater DXA whole body fat percent, with the explained variance increasing by 1.5% compared to model 1 (Table 3). For females, the unstandardized β coefficient (95%CI) for log-CRP for model 2 was 0.89 (0.80, 0.98) and the standardized β coefficient (95%CI) was 0.14 (0.13, 0.16) (Table 3), indicating a small effect size [39]. When the interaction term (log-CRP×subgroup) was added to model 2, there was no meaningful change in the explained variance of the model (model 3) (Table 2).

For males, model 1 explained 54.9% of the variance in DXA whole body fat percent (Table 2). When log-CRP was added to model 1, there was a meaningful increase in the explained variance of the model (model 2) (Table 2). Specifically, after adjusting for multiple comparisons, a greater log-CRP level was associated with greater DXA whole body fat percent, with the explained variance increasing by 1.6% compared to model 1 (Table 3). For males, the unstandardized β coefficient (95%CI) for log-CRP for model 2 was 0.77 (0.69, 0.85) and standardized β coefficient (95%CI) was 0.13 (0.12,

**Table 4. Unstandardized and standardized β coefficients (95%CI) for log-CRP for model 3 (covariates†+log-CRP+log-CRP×subgroup) for females and males.**

| | Model 3 *Unstandardized* log-CRP β Coefficient (95%CI) | Model 3 *Standardized* log-CRP β Coefficient (95%CI) |
|---|---|---|
| **Females** | | |
| Grip Strength (kg) | −0.54 (−0.72, −0.36)‡ | −0.09 (−0.11, −0.07)‡ |
| Mobility Index | −0.05 (−0.12, 0.02)‡ | −0.05 (−0.07, −0.03)‡ |
| DXA Whole Body Fat Mass (g) | 664.07 (539.78, 788.36)‡ | 0.05 (0.04, 0.06)‡ |
| DXA Whole Body Fat Percent (%) | 1.22 (1.09, 1.36)‡ | 0.15 (0.13, 0.16)‡ |
| DXA Whole Body Lean Mass (g) | −628.15 (−780.29, −476.01)‡ | −0.08 (−0.10, −0.07)‡ |
| DXA Trunk Fat Percent (%) | 1.66 (1.50, 1.82)‡ | 0.17 (0.15, 0.18)‡ |
| DXA Appendicular Fat Mass Index (kg/m$^2$) | 0.05 (0.03, 0.08)‡ | 0.02 (0.01, 0.03)‡ |
| DXA Appendicular Lean Mass Index (kg/m$^2$) | −0.12 (−0.14, −0.10)‡ | −0.11 (−0.12, −0.09)‡ |
| **Males** | | |
| Grip Strength (kg) | −0.64 (−0.91, −0.36)‡ | −0.08 (−0.10, −0.06)‡ |
| Mobility Index | −0.03 (−0.11, 0.04)‡ | −0.05 (−0.06, −0.03)‡ |
| DXA Whole Body Fat Mass (g) | 584.06 (457.82, 710.30)‡ | 0.07 (0.06, 0.08)‡ |
| DXA Whole Body Fat Percent (%) | 0.85 (0.73, 0.97)‡ | 0.13 (0.12, 0.15)‡ |
| DXA Whole Body Lean Mass (g) | −554.95 (−744.14, −365.76)‡ | −0.07 (−0.08, −0.05)‡ |
| DXA Trunk Fat Percent (%) | 1.15 (1.02, 1.29)‡ | 0.14 (0.12, 0.15)‡ |
| DXA Appendicular Fat Mass Index (kg/m$^2$) | 0.04 (0.02, 0.05)‡ | 0.05 (0.04, 0.06)‡ |
| DXA Appendicular Lean Mass Index (kg/m$^2$) | −0.12 (−0.14, −0.10)‡ | −0.12 (−0.13, −0.11)‡ |

Note. *CRP*, C-reactive protein; *DXA*, Dual-Energy X-Ray Absorptiometry.

†Covariates included *age*, *body mass index*, *education level*, *income level*, *social inequality*, *scores on the CES-D* and *subgroup*.

‡Indicates a Benjamini-Hochberg correction for multiple comparisons was applied.

0.15) (Table 3), indicating a small effect size [39]. When the interaction term (log-CRP×subgroup) was added to model 2, there was no meaningful change in the explained variance of the model (model 3) (Table 2).

**DXA trunk fat percent.** For females, model 1 (covariates only) explained 60.4% of the variance in DXA trunk fat percent (Table 2). When log-CRP was added to model 1, there was a meaningful increase in the explained variance of the model (model 2) (Table 2). Specifically, after adjusting for multiple comparisons, a greater log-CRP level was associated with greater DXA trunk fat percent, with the explained variance increasing by 1.9% (Table 3). For females, the unstandardized β coefficient (95%CI) for log-CRP for model 2 was 1.23 (1.12, 1.34) and the standardized β coefficient (95%CI) was 0.16 (0.15, 0.18) (Table 3), indicating a small effect size [39]. When the interaction term (log-CRP×subgroup) was added to model 2, there was no meaningful change in the explained variance of the model (model 3) (Table 2).

For males, model 1 explained 56.2% of the variance in DXA trunk fat percent (Table 2). When log-CRP was added to model 1, there was a meaningful increase in the explained variance of the model (model 2) (Table 2). Specifically, after adjusting for multiple comparisons, a greater log-CRP level was associated with greater DXA trunk fat percent, with the explained variance increasing by 1.6% (Table 3). For males, the unstandardized β coefficient (95%CI) for log-CRP for model 2 was 0.93 (0.83, 1.02) and standardized β coefficient (95%CI) was 0.13 (0.12, 0.15) (Table 3), indicating a small effect size [39]. When the interaction term (log-CRP×subgroup) was added to model 2, there was no meaningful change in the explained variance of the model (model 3) (Table 2).

**DXA appendicular lean mass index.** For females, there was no meaningful change in the explained variance of the model (represented by an increase in the adjusted $R^2$ by 1% or more) with the addition of log-CRP (model 2) or with the addition of the interaction term (log-CRP×subgroup) to the regression (model 3) (Table 2).

**Table 5.** Unstandardized and standardized β coefficients (95%CI) for each covariate for females and males with DXA whole-body fat percent as the dependent variable.

| Females | *Unstandardized* β Coefficient (95%CI) | *Standardized* β Coefficient (95%CI) |
|---|---|---|
| Age (yrs) | 0.09 (0.08, 0.10)‡ | 0.13 (0.12, 0.15)‡ |
| BMI (kg/m²) | 0.77 (0.76, 0.79)‡ | 0.74 (0.73, 0.75)‡ |
| Education Level | | |
| 1 | – | – |
| 2 | 0.27 (−0.09, 0.63)‡ | 0.01 (−0.00, 0.03)‡ |
| 3 | 0.16 (−0.15, 0.48)‡ | 0.01 (−0.01, 0.03)‡ |
| 4 | −0.04 (−0.47, −0.39)‡ | −0.00 (−0.02, 0.01)‡ |
| 5 | −0.28 (−0.59, 0.03)‡ | −0.02 (−0.04, 0.00)‡ |
| 6 | −0.57 (−0.90, −0.24)‡ | −0.04 (−0.06, −0.02)‡ |
| 7 | −0.87 (−2.88, 1.14)‡ | −0.01 (−0.02, 0.01)‡ |
| Income Level | | |
| 1 | – | – |
| 2 | 0.34 (−0.07, 0.75)‡ | 0.02 (−0.00, 0.05)‡ |
| 3 | 0.15 (−0.25, 0.55)‡ | 0.01 (−0.02, 0.04)‡ |
| 4 | −0.03 (−0.46, 0.39)‡ | −0.00 (−0.03, 0.03)‡ |
| 5 | −0.77 (−1.21, −0.32)‡ | −0.05 (−0.07, −0.02)‡ |
| Social Inequality | 0.01 (−0.04, 0.05)‡ | 0.00 (−0.01, 0.02)‡ |
| CES-D Scores | 0.03 (0.01, 0.05)‡ | 0.02 (0.01, 0.04)‡ |
| Subgroup | | |
| A | – | – |
| B | 0.16 (−0.04, 0.37)‡ | 0.01 (−0.00, 0.03)‡ |
| C | 0.17 (−0.18, 0.51)‡ | 0.01 (−0.01, 0.02)‡ |
| D | −0.30 (−0.63, 0.03)‡ | −0.01 (−0.03, 0.00)‡ |
| E | −0.19 (−0.61, 0.24)‡ | −0.01 (−0.02, 0.01)‡ |
| F | −0.14 (−0.55, 0.26)‡ | −0.00 (−0.02, 0.01)‡ |
| G | −0.45 (−0.95, 0.04)‡ | −0.01 (−0.03, 0.00)‡ |
| H | −0.13 (−0.51, 0.25)‡ | −0.00 (−0.02, 0.01)‡ |
| **Males** | *Unstandardized* β Coefficient (95%CI) | *Standardized* β Coefficient (95%CI) |
| Age (yrs) | 0.10 (0.09, 0.11)‡ | 0.18 (0.16, 0.19)‡ |
| BMI (kg/m²) | 0.83 (0.82, 0.85)‡ | 0.71 (0.70, 0.72)‡ |
| Education Level | | |
| 1 | – | – |
| 2 | −0.31 (−0.63, −0.00)‡ | −0.02 (−0.04, −0.00)‡ |
| 3 | −0.06 (−0.37, 0.24)‡ | −0.00 (−0.02, 0.02)‡ |
| 4 | 0.10 (−0.33, 0.53)‡ | 0.00 (−0.01, 0.02)‡ |
| 5 | −0.38 (−0.67, −0.10)‡ | −0.03 (−0.06, −0.01)‡ |
| 6 | 0.21 (−0.07, 0.50)‡ | 0.02 (−0.01, 0.04)‡ |
| 7 | −0.43 (−2.17, 1.31)‡ | −0.00 (−0.02, 0.01)‡ |
| Income Level | | |
| 1 | – | – |
| 2 | −0.50 (−0.97, −0.04)‡ | −0.03 (−0.06, −0.00)‡ |
| 3 | −0.82 (−1.27, −0.37)‡ | −0.07 (−0.11, −0.03)‡ |
| 4 | −1.13 (−1.59, −0.67)‡ | −0.09 (−0.13, −0.05)‡ |
| 5 | −1.41 (−1.87, −0.94)‡ | −0.11 (−0.15, −0.07)‡ |

*(Continued)*

**Table 5.** (Continued)

| Females | *Unstandardized* β Coefficient (95%CI) | *Standardized* β Coefficient (95%CI) |
|---|---|---|
| Social Inequality | −0.04 (−0.08, −0.00)‡ | −0.01 (−0.03, −0.00)‡ |
| CES-D Scores | 0.03 (0.02, 0.05)‡ | 0.03 (0.01, 0.04)‡ |
| Subgroup | | |
| A | – | – |
| B | 0.55 (0.38, 0.72)‡ | 0.05 (0.03, 0.06)‡ |
| C | −0.47 (−0.79, −0.14)‡ | −0.02 (−0.03, −0.01)‡ |
| D | 0.27 (−0.03, 0.57)‡ | 0.01 (−0.00, 0.03)‡ |
| E | 0.03 (−0.52, 0.57)‡ | 0.00 (−0.01, 0.01)‡ |
| F | 0.61 (0.14, 1.07)‡ | 0.02 (0.00, 0.03)‡ |
| G | −0.37 (−1.07, 0.34)‡ | −0.01 (−0.02, 0.01)‡ |
| H | −0.07 (−0.56, 0.41)‡ | −0.00 (−0.02, 0.01)‡ |

Note. *DXA*, Dual-Energy X-Ray Absorptiometry. *BMI*, body mass index. *CES-D*, Center for Epidemiological Studies Short Depression scale.

‡Indicates a Benjamini-Hochberg correction for multiple comparisons was applied.

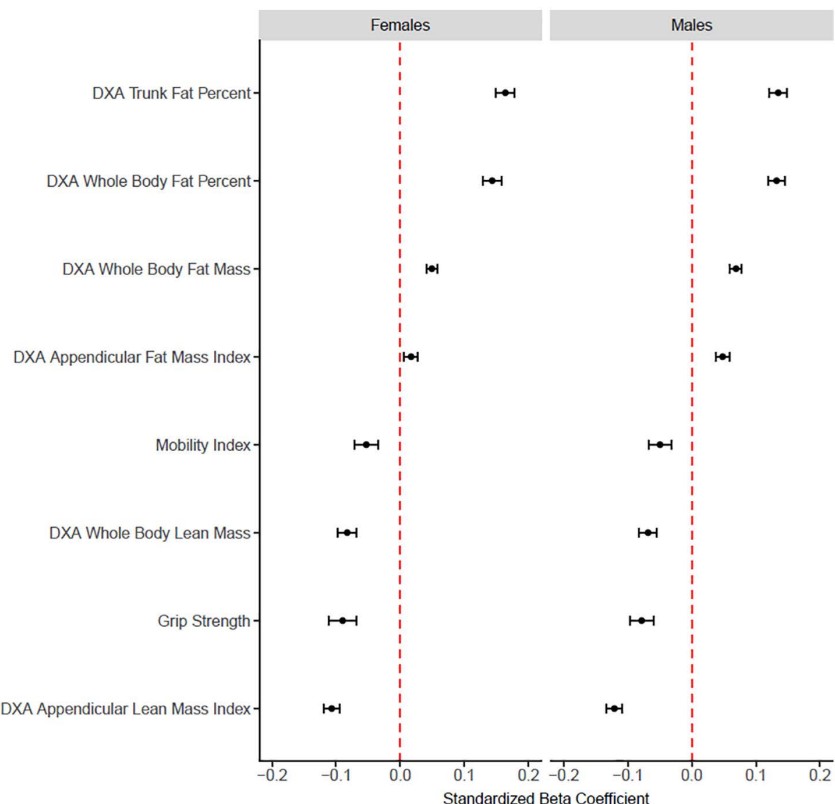

**Fig 4. Plot comparing the standardized β coefficients (95%CI) of the associations between dependent variables reflecting body composition and physical function with the independent variable of log-CRP for model 2 (covariates + log-CRP) for females and males.**

For males, model 1 (covariates only) explained 65.6% of the variance in DXA appendicular lean mass index (Table 2). When log-CRP was added to model 1, there was a meaningful change in the explained variance of the model (model 2) (Table 2). Specifically, after adjusting for multiple comparisons, a greater log-CRP level was associated with reduced DXA appendicular lean mass index, with the explained variance increasing by 1.2% (Table 3). For males, the unstandardized β coefficient (95%CI) for log-CRP for model 2 was −0.14 (−0.16, −0.13) and the standardized β coefficient (95%CI) was −0.12 (−0.13, −0.11) (Table 3), indicating a small effect size [39]. When the interaction term (log-CRP × subgroup) was added to model 2, there was no meaningful change in the explained variance of the model (model 3) (Table 2).

## Discussion

In this large sample of Canadian adults (45–85 years), in both females and males and in those with and without OA, greater adiposity represented by DXA whole body and trunk fat percent was associated with greater systemic inflammation. Over and above covariates, log-CRP increased the explained variance in whole body and trunk fat percent by 1.5% and 1.9%, respectively for females and 1.6% and 1.6%, respectively for males. For females and males, a one unit increase in log-CRP was associated with a 0.89% and 0.77% increase in DXA whole body fat percent, respectively; and a 1.23% and 0.93% increase in DXA trunk fat percent, respectively. Among males, lower DXA appendicular lean mass index, reflecting less muscle in the arms and legs relative to height, was also linked with greater systemic inflammation. Therefore, the patterns observed in associations between CRP and measures of functional capacity and body composition were similar in females and males, with the single exception indicating that, in males, appendicular lean mass index was inversely associated with systemic inflammation. These associations were also not altered by variations in subgroup status; thus, we did not observe a stronger (or weaker) association of CRP in the presence of multimorbidities or OA. However, for the remaining dependent variables (grip strength, mobility index, DXA whole body fat mass, DXA whole body lean mass, DXA appendicular fat mass index), there was no meaningful association with log-CRP. The addition of the interaction term (log-CRP × subgroup) did not meaningfully contribute to the models for any of the dependent variables. Overall, these results show that the percentage of whole-body and trunk adipose tissue is related to the degree of systemic inflammation among Canadian adults, regardless of the accumulation of chronic health conditions, or the presence of OA. However, it should be noted that the effect size is small (Table 3) and this likely indicates that the influence of systemic inflammation on the development and progression of OA is only a small part of a complex system of interconnected factors.

In this large, cross-sectional analysis of Canadians aged 45–85 years, central adiposity but not OA disease or multimorbidity were related with systemic inflammation. This greater central adiposity likely reflects greater visceral adiposity which is directly associated with increased cardiovascular and diabetes risk [40]. This greater risk for chronic cardiometabolic conditions could be explained by our observation that accumulation of fatty tissue in the whole body and trunk was linked to higher levels of serum CRP. In general, fat mass percentage is higher in individuals with OA compared to controls [41]. This adiposity likely worsens OA disease through multiple mechanisms, such as insulin resistance, altered immune response and inflammatory cells, degradation of cartilage [10] and reduced physical performance in those with OA [42]. Higher body fat percentage is also associated with reduced physical function in individuals with multimorbidities [42]. However, in the current analysis, there was no association of higher CRP levels with DXA whole body fat mass (as opposed to whole body fat percent). Interestingly, there are multiple instances where fat percent is the only body composition measure that is significantly associated with CRP [43–45] or disability [42,46] in individuals with multiple sclerosis [43], obesity [44], metabolic syndrome [45] and knee OA [42,46]. These findings indicate that, within a cross-sectional analysis (Canadians aged 45–85 years), there exists no specific absolute volume of fat that would necessarily link with systemic inflammatory processes. Instead, based on the current results and previous studies [43–45], the relative amount of fat compared to other body tissues may be a better indicator of low-grade inflammation and centralized adiposity that appears in chronic conditions, including but not exclusive to OA. Overall, greater levels of circulating inflammatory

mediators, reflected by greater adipose tissue, may be a key contributor to OA worsening through different inflammatory processes, including pathways signalled by higher CRP levels.

There was no association between the physical capacity measures and CRP. Previous cross-sectional and longitudinal analyses with smaller samples have also found no relationship between baseline CRP and functional outcomes, including gait speed [47], knee extensor strength [47] and change in standing and walking functional capacity measures [48]. However, for individuals who experience a high CRP level over 10 years, there is an increased risk of deterioration in activities of daily living, balance and walking speed [49] and higher CRP was associated with reduced gait speed [50]. It should be noted that previous analyses included participants that were in good physical and mental health [48], excluded participants using anabolic steroids and/or corticosteroids [47] and excluded participants who had higher CRP levels (>10 mg/L) [49]. Those with higher CRP levels who were excluded were more likely to have lower education, currently smoking, hypertensive, less physically active and report greater limitations with activities of daily living [49]. Therefore, participants in previous analyses likely had fewer multimorbidities than participants in the current analysis. Rather than systemic inflammation, physical capacity outcomes are likely better related to physical activity engagement, willpower beliefs and overall health demands [51].

The lack of association of log-CRP, even among subgroups with multimorbidities, was unexpected because the level of CRP is proportional to the intensity of inflammation [26] and greater CRP levels are associated with many chronic health conditions [18–25]. However, there are a number of reasons for this observation. First, multimorbidity was defined as a binary variable and derived from a large variety of chronic conditions. Second, there may exist considerable variability at any single point in time that results from a wide variety of multimorbidities. The time-course of some chronic conditions included in this analysis (e.g., rheumatoid arthritis) may not be steady, which may lead to fluctuating levels of CRP over time. CRP is regulated by other pro-inflammatory cytokines, including interleukin-6 and tumor necrosis factor-alpha, which are secreted by various inflammatory and immunological cells [52]. Future studies would benefit from exploring these pro-inflammatory mediators to better determine the mechanisms underlying the association between OA, multimorbidity, body composition and physical capacity. Third, subgroup as a variable, was defined according to a combination of hand OA, lower extremity OA and multimorbidity status. We were particularly interested in the interaction between CRP and subgroup. As a result however, the association with multimorbidity was not directly measured. Fourth, self-report may also confound the associations. On the other hand, the percent fat may have a more 'consistent' relationship with CRP, regardless of the specific chronic condition and its underlying inflammatory process. The relationship between circulating inflammation and fat accumulation appears strong, consistent and well-established [12,15–17,53]. Since adipose tissue is associated with higher levels of circulating inflammatory mediators, including CRP [12,15–17], body composition appears to have a unique influence on CRP [18].

There was no relationship observed between log-CRP and DXA whole body lean mass. This finding is inconsistent with previous studies that found an association between higher levels of CRP and lower lean mass [25,54]. However, for males, there was a relationship between log-CRP and DXA appendicular lean mass index, suggesting sex-related differences in appendicular lean mass changes. It has been suggested that loss of lean mass occurs, in part, as a result of the catabolic effects of inflammation on muscle tissue, which ultimately leads to a decline in physical function [55,56]. Further, there was no meaningful association between CRP and any of the functional outcomes (grip strength, mobility index) measured in the current study. Previous work shows that there is a weaker relationship of physical function with lean mass than measures of the accumulation of fat [57,58]. Thus, loss of muscle mass and function in those with OA and other multimorbidities may not be directly or entirely related to the degree of systemic inflammation.

## Limitations and future directions

We analyzed data previously collected through the CLSA. Although the CLSA is transparent with their data collection methods, we were not able to directly collect the data and thus, are somewhat removed from verifying the accuracy. Additionally,

the CLSA is limited to Canadian adults aged 45–85 years. Therefore, the generalizability of the dataset and any results derived from such data is limited to similar populations. Limitations associated with the statistical modelling included the (i) definition of multimorbidity as a binary variable (presence or absence); (ii) inclusion of subgroup as a covariate rather than separate analyses per group; (iii) criteria for statistical significance (adjusted $R^2$ value increasing by 1% or more), and (iv) use of cross-sectional data that limits the ability to infer causality. More sophisticated analyses that reflect the relative weight of different chronic health conditions and their association with CRP might yield different results, however there is no consensus on a list of chronic health conditions that are particularly associated with systemic inflammation. Additionally, distinguishing between the type of adipose tissue (e.g., visceral, subcutaneous, etc.) and its impact on inflammation would also be beneficial, although beyond the scope of the current study. Several outcomes, including the presence of OA and other health conditions are self-reported, which may incorporate biases (e.g., social desirability); but these self-report outcomes reflect the participants' understanding of their health status and are common methods to collect data in a large-scale epidemiological study. Additionally, CRP provides one measure of inflammation. CRP is not the only marker of inflammation. It is possible that OA is associated with other inflammatory responses not measured in the current study. As such, the CLSA provides a unique and powerful opportunity to use multiple different measures from a very large cohort.

Although a large sample size is a strength of the current analysis, it posed a unique challenge. Likely due to the large sample size, the addition of our independent variable (CRP) and interaction term resulted in statistically significant associations for many of the outcomes, despite the explained variance increasing by < 1% for 5 of the 8 dependent measures. It is important to consider whether these associations identify meaningful relationships between the dependent variables and CRP. To directly address this challenge, we predefined that a 1% increase in explained variance was necessary to be considered meaningful, in terms of effect size.

Future directions might explore whether the proportion of body tissue attributed to fat might be a valuable clinical outcome for the conservative treatment of inflammatory chronic diseases such as OA. Additionally, exploring other markers of inflammation, such as interleukin-6 and tumour necrosis factor may be worthwhile to explore in OA to reflect different inflammatory pathways. Finally, an analysis involving a greater number of subgroups that further partition chronic conditions may provide a better opportunity to identify the nuances associated with markers of inflammation and their role in obesity-related multimorbidities.

## Conclusion

In a large sample of Canadian adults, a higher percent fat was associated with higher levels of systemic inflammation, as measured by CRP. This relationship is consistent regardless of multimorbidity status, sex and the presence of OA. Inflammation may be the common element between obesity and the presence of many chronic health conditions that occur with aging, including OA. Although future work is required to better elucidate the relationship between inflammation and the development of OA, the results of the present study highlight the important role of adipose tissue in the inflammatory process. Combined with prior studies on obesity and OA, it may be suggested that OA is an inflammatory disease, in part a consequence of excessive adipose tissue over the long-term.

## Supporting information

**S1 File. Canadian Longitudinal Study on Aging (CLSA) Standard Operating Procedures (SOPs) and Questionnaires.**
(DOCX)

**S2 File. Supplementary Table 1. Demographic characteristics and baseline values for participants included in the statistical analyses for each of the 8 dependent variables.**
(DOCX)

**S3 File. Supplementary Table 2. Number (n) of participants for each dependent variable (grip strength, mobility index, DXA whole body fat mass, DXA whole body fat percent, DXA whole body lean mass, DXA trunk fat percent, DXA appendicular fat mass index, DXA appendicular lean mass index), separated by sex (females, males) and subgroup (A through H).**
(DOCX)

## Acknowledgments

The authors gratefully acknowledge the time and commitment of the CLSA participants, without whom this research would not be possible.

The opinions expressed in this manuscript are the author's own and do not reflect the views of the Canadian Longitudinal Study on Aging (CLSA).

## Author contributions

**Conceptualization:** Kendal A. Marriott, Paul W. Stratford, Chris P. Verschoor, Dawn M.E. Bowdish, Marina Mourtzakis, Jaclyn N. Chopp-Hurley, Emily G. Wiebenga, Monica R. Maly.

**Formal analysis:** Kendal A. Marriott.

**Funding acquisition:** Monica R. Maly.

**Methodology:** Kendal A. Marriott, Paul W. Stratford, Chris P. Verschoor, Dawn M.E. Bowdish, Marina Mourtzakis, Jaclyn N. Chopp-Hurley, Monica R. Maly.

**Supervision:** Monica R. Maly.

**Writing – original draft:** Kendal A. Marriott.

**Writing – review & editing:** Kendal A. Marriott, Paul W. Stratford, Chris P. Verschoor, Dawn M.E. Bowdish, Marina Mourtzakis, Jaclyn N. Chopp-Hurley, Emily G. Wiebenga, Monica R. Maly.

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
