## [Decision Letter · Decision Letter 0]

3 Sep 2024

Dear Dr. Maly,

When you have completed the revisions, please submit your revised manuscript by Oct 18 2024 11:59PM. If you will need more time than this to complete your revisions, please reply to this message or contact the journal office at plosone@plos.org . A rebuttal letter that responds to each point raised by the academic editor and reviewer(s). You should upload this letter as a separate file labeled 'Response to Reviewers'.A marked-up copy of your manuscript that highlights changes made to the original version. You should upload this as a separate file labeled 'Revised Manuscript with Track Changes'.An unmarked version of your revised paper without tracked changes. You should upload this as a separate file labeled 'Manuscript'.

We look forward to receiving your revised manuscript.

Kind regards,

Stephen E Alway, Ph.D.

Academic Editor

PLOS ONE

“This research was made possible using the data/biospecimens collected by the Canadian Longitudinal Study of Aging (CLSA). Funding for the CLSA is provided by the Government of Canada, through the Canadian Institutes of Health Research (CIHR) under grant reference [LSA 94473] and the Canada Foundation for Innovation, as well as the following provinces, Newfoundland, Nova Scotia, Quebec, Ontario, Manitoba, Alberta, and British Columbia. This research has been conducted using the CLSA Baseline Comprehensive Dataset 5.1, under Application Number 2002013. The CLSA is led by Drs. Parminder Raina, Christina Wolfson and Susan Kirkland.”

“The authors disclosed receipt of the following financial support for the research, authorship, and/or publication of this article: This research was made possible using the data/biospecimens collected by the Canadian Longitudinal Study of Aging (CLSA). Funding for the CLSA is provided by the Government of Canada, through the Canadian Institutes of Health Research (CIHR) under grant reference [LSA 94473] and the Canada Foundation for Innovation, as well as the following provinces, Newfoundland, Nova Scotia, Quebec, Ontario, Manitoba, Alberta, and British Columbia. This research has been conducted using the CLSA Baseline Comprehensive Dataset 5.1, under Application Number 2002013. The CLSA is led by Drs. Parminder Raina, Christina Wolfson and Susan Kirkland. This work was supported by the Arthritis Society Postdoctoral Fellowship Salary Award [award #0000000039] to [KAM]; Canada Research Chair in Aging and Immunity to [DMEB]; Canadian Institutes of Health Research (CIHR) Fellowship Award to [JC-H]; and The Arthritis Society Stars Mid-Career Development Award funded by the Canadian Institutes of Health Research-Institute of Musculoskeletal Health and Arthritis to [MRM]. The funders had no role in study design, data collection and analysis, decision to publish, or preparation of the manuscript.”

Reviewers' comments:

Reviewer's Responses to Questions

**Comments to the Author**

1. Is the manuscript technically sound, and do the data support the conclusions?

Reviewer #1: Yes

Reviewer #2: Yes

2. Has the statistical analysis been performed appropriately and rigorously?

Reviewer #1: I Don't Know

Reviewer #2: Yes

3. Have the authors made all data underlying the findings in their manuscript fully available?

Reviewer #1: No

Reviewer #2: Yes

4. Is the manuscript presented in an intelligible fashion and written in standard English?

Reviewer #1: Yes

Reviewer #2: Yes

Reviewer #1: Summary

This study aims to explore the association between C-reactive protein (CRP) and physical function and body composition in CLSA participants with/without osteoarthritis (OA) and multimorbidity (MM). The analyses are disaggregated by sex, and linear regression is used where three models are compared: model 1 (covariates only, including subgroup defined by OA and MM categories), model 2 (covariates + log-CRP), and model 3 (covariates + log-CRP + log CRP x subgroup interaction). A change of 1%+ in R2 across the models is used to identify significance. Percentage adiposity measured by whole body fat percentage and trunk fat percentage was associated with elevated CRP (small effect size) in both females and males. Lower appendicular lean body mass was associated with elevated CRP (small effect size) in males only. These results were observed regardless of OA status, suggesting that percent adiposity and not OA is associated with systemic inflammation (measured by CRP).

Overall, this study is well done and the findings are quite interesting. To strengthen this paper, some of the

methodological choices require further clarification/justification, and should as well be acknowledged as limitations, as noted in the detailed comments below.

Detailed Comments:

1. Abstract:

i. Objectives: Since all participants in the CLSA comprehensive database were included (complete cases) and the age range starts at 45 years, it may be misleading to refer to the participants in this study as ‘older’. It would be better to either state the age range explicitly (45-85 years), or remove ‘older’ as a qualifier.

ii. Methods: It should be stated that analyses were sex disaggregated, as some results (appendicular lean mass index) held for only one sex (males). Also, the criterion for deciding statistical significance (change in adjusted R-squared of 1%+ when CRP added to model with only covariates) should be included in the methods, as the results and interpretations are highly dependent on this choice (as noted in the manuscript, many more statistically-significant results were achieved [driven by the large sample)], but did not reach this threshold).

iii. Conclusions: It is stated that percentage adiposity and appendicular lean mass index were associated with CRP regardless of OA status. Given that the subgroups were defined by OA and MM status, what happened to MM? MM seems to be presented sporadically, which is confusing – e.g., MM is referred to in the title, referred to in the Methods and the Results, but not in the Conclusions. The manuscript reads this way as well – e.g., MM is not discussed in the Introduction, yet it appears later in the manuscript (Participant & Discussion, see comments below). The role of MM in this study is unclear and inconsistently presented.

2. Introduction:

i. Osteoarthritis (OA) and its linkages with obesity, systemic inflammation (measured by CRP), and physical function/mobility/physical activity appear to be the primary focus, as the current evidence and outstanding questions focus on these topics. There is no discussion of multimorbidity (MM) and it is unclear what role it plays in this study; the evidence linking MM with pain and physical function (in those with lower extremity OA) is only briefly mentioned in the Participants section of the paper and in the Discussion (where the evidence of the links between chronic conditions and inflammation are discussed). All of this evidence of MM should be moved to the Introduction, including the evidence linking chronic conditions (and by extension MM) to systemic inflammation.

3. Methods:

i. Participants:

i. The analyses were done for complete cases. Please indicate the portion of the 30,000 comprehensive cohort participants that this represents (or the % of missing data pertaining to this cohort).

ii. Subgroups: A categorical variable is included among the covariates that has eight categories for the different combinations of OA and MM. MM is reflected as a binary variable (Yes if 2+ chronic conditions, No if < 2 chronic conditions), yet the stated evidence linking MM with pain and physical function pertains to the number of MM. Another way to model MM is to have this as a separate covariate with the measure being the number of chronic conditions (e.g., aligned with the stated evidence, increased sensitivity to participant differences in MM), and to have OA as its own (categorical) variable. Would this approach be a better way to capture MM (and OA) in the model?

iii. Data and Statistical Analysis:

1. A change in the adjusted R2 of 1%+ is used to identify a significant result (due to the large sample sizes and resulting small p-values). While using the change in the adjusted R2 is a common approach to this problem, the choice of the threshold (1%+) requires justification; the results/interpretations are highly impacted by this choice, thus the rationale or supporting reference(s) for this threshold should be provided.

2. An alternate modelling approach is to perform a disaggregated analysis by subgroup and sex. This would bring the sample sizes down considerably, yet still provide a sufficient sample for the regression analyses (according to the sample sizes for the sex/subgroup cells shown in the S3 supplement). Perhaps this would enable use of more traditional statistical significance criteria (p-value versus the change in adjusted R2 threshold which is difficult to set)? Does this warrant consideration?

4. Results:

i. Model Outcomes: Unstandardized and standardized betas (#’s, 95% CIs, figure) are provided in the main body of the manuscript for the linear regressions run for each of the eight dependent variables. It would be helpful to include the results from S5 in the main body of the manuscript. This explains the focus on the results (on only those achieving significance), and provides a sense of the statistical significance (or insignificance) and change in magnitude of the adjusted R2 across the eight outcomes.

ii. Additional Supplement: It would also be helpful to provide the full set of results for the linear regression models, that includes the results for the covariates. In particular, the subgroup results may be interesting to readers, as OA is discussed at length in the Introduction (and the subgroups are defined by the OA groups). This may also allow readers to assess the extent to which some of the hypothesized results emerged in the models (even if not significant according to the R2 threshold)– e.g., is CRP more strongly associated with physical function in those with lower extremity OA compared to hand OA?

5. Discussion:

i. It is stated that the “lack of association of log-CRP with the number of multimorbidities as reflected by subgroup analyses was unexpected”, and that this contradicts findings in the literature relating CRP with inflammation and inflammation with many health conditions. This statement should be revised, because number of multimorbidities was not modelled directly in this study, it was captured in a subgroup where MM was defined as a binary variable. This modelling choice could account for the divergence of this study’s findings with the broader literature.

6. Limitations and Future Directions:

i. Limitations should be revised to include some of the modelling choices made in this study (e.g., MM defined as a binary variable, subgroup captured as a covariate versus disaggregation by subgroup) and criterion set for statistical significance (1%+ threshold for change in R2).

7. Conclusion:

i. The second sentence states that the relationship is consistent “regardless of the number of chronic conditions”. This should be rephrased to state that the relationship is consistent” regardless of MM status” (number of chronic conditions was not captured in the analysis, only MM defined as a binary variable).

Reviewer #2: In this manuscript, the authors seek to understand associations between serum inflammation (as measured via C-reactive protein, body composition, and physical capacity among Canadian adults with and without Osteoarthritis and/or multimorbidity. Following their analyses, they find that CRP levels are associated with body composition, regardless of OA or MM status, and some sex differences can also be observed. The paper is well written and very detailed. Additionally, it benefits from using a large accessible data set and study that used gold-standard methods for assessment of body composition (DXA). However, there are several revisions requested before the paper can move forward.

1) It is recommended that greater detail and transparency be presented in the results and discussion regarding the associations measured between CRP and the subgroups, to better integrate the sections of the paper. There is a large focus on the associations between inflammation and OA, and multimorbidity in the introduction of the paper. However, these results get somewhat lost in the results section. The authors acknowledge they are not statistically significant, but without being able to view the models that include the subgroups, one cannot adjudicate whether they may indicate some degree of biological significance. Additionally, as the discussion delves into the observations that associations between degrees/types of OA and multimorbidity were negligible or absent, this should be more apparent in the results tables and text. The paper would also benefit from greater discussion regarding the potential reasons why the results here don't necessary reflect what has been found elsewhere.

2) The paper would benefit from removing some of the tables from the supporting information section, and moving them to the results. As presented, the results are difficult to follow given that many of the results are presented in the supporting information section.

3) As stated, the strongest associations were between central adiposity and CRP. Greater discussion of what may be causing that would benefit the paper. For example, does that association indicate that visceral adiposity (known to be strongly associated with pro-inflammatory cytokines) is contributing to these observations?

4) the paper may also benefit from including additional future directions, such as recommendations for different types of analyses or ways to investigate that may better assess the relationship between adiposity, inflammation, and OA/multimorbidity.

The following minor revisions are also requested.

Page 3 Line 57 – 58: The phrasing is a bit redundant. Meaningful engagement is loaded/subjective and may need defining. Otherwise, I recommend replacing meaningful with 'typical'.

Page 3, line 61: add ‘and’ before altered

Page 3, line 65 – 67: This is a general; it would benefit from specificity (e.g., give examples of which pro-inflammatory cytokines).

Page 4, line 87: comma after hand OA

Page 4, line 88 – 89: better is subjective- replace with the specifics, (e.g., lower body fat %, lower BMI, etc.), and 'improved physical capacity"

Page 4, line 90: This sentence needs clarification. Following the prior sentence, the assumption is made that the hypothesis is that lower levels of CRP have a stronger association with OA compared to those without OA, etc.

Page 5, line 114-115: Evaluate the tense of access here

Page 7, line 162: is this hsCRP?

Page 13, line 287 – 289: While the explanations are great, some of the detail isn't necessary. You should assume your reader will know that a CI that doesn't cross 0 is significant.

Page 16: Results: As the paper is focused heavily on OA, there should be greater inclusion of results regarding that variable.

Page 17, lines 372 – 374: There needs to be greater delineation in the results section of how these levels of systemic inflammation are occurring regardless of OA or multimorbidity.

Page 18, line 387 – 388: It may be beneficial to also discuss types of fat measured (e.g., subcutaneous vs. visceral) and how that may impact inflammation levels.

Page 18, lines 391- 394: This isn't the focus of the paper; associations between CRP and more severe OA are not clear in the results presented.

Page 19, line 429: As the paper isn't assessing sarcopenic changes, this is speculative.

Page 20, lines 446 – 447: As CRP increases with acute infection, was this controlled for?

**Do you want your identity to be public for this peer review?** For information about this choice, including consent withdrawal, please see our Privacy Policy

Reviewer #1: No

Reviewer #2: No

---

## [Author Response · Author response to Decision Letter 1]

18 Oct 2024

Editor Comments: Please see cover letter attached.

Reviewer Comments: Please see "Response to Reviewers" document attached.

---

## [Decision Letter · Decision Letter 1]

11 Jul 2025

Dear Dr. Maly,

Thank you for submitting your manuscript to PLOS ONE. After careful consideration, we feel that it has merit but does not fully meet PLOS ONE’s publication criteria as it currently stands. Therefore, we invite you to submit a revised version of the manuscript that addresses the points raised during the review process.

You’ve addressed most of the reviewers’ comments thoroughly, and they were generally satisfied with the revisions. However, since both recommended a revision for minor clarifications, I am issuing a **Minor Revisio****n** decision to allow you to address their final comments before acceptance. Please submit an updated manuscript with a point-by-point response.

We look forward to receiving your revised manuscript.

Kind regards,

Zohreh Sajadi Hezaveh

Academic Editor

PLOS ONE

Journal Requirements:

Reviewers' comments:

Reviewer's Responses to Questions

**Comments to the Author**

Reviewer #2: All comments have been addressed

Reviewer #3: (No Response)

Reviewer #4: (No Response)

2. Is the manuscript technically sound, and do the data support the conclusions?

Reviewer #2: Yes

Reviewer #3: Yes

Reviewer #4: Yes

3. Has the statistical analysis been performed appropriately and rigorously?

Reviewer #2: Yes

Reviewer #3: Yes

Reviewer #4: Yes

4. Have the authors made all data underlying the findings in their manuscript fully available?

Reviewer #2: Yes

Reviewer #3: Yes

Reviewer #4: Yes

5. Is the manuscript presented in an intelligible fashion and written in standard English?

Reviewer #2: Yes

Reviewer #3: Yes

Reviewer #4: Yes

Reviewer #2: The authors did well to address all prior requested revisions. The revised paper is well organized and presents interesting insights into associations between adiposity, OA, and other multi-morbidities. The paper is now more concise and direct, and the inclusion of more tables/figures into the main text facilitates better understanding of the findings.

A few very minor remaining recommendations 1) Page 9 line 207, please move up the definition of DXA before using the acronym in '...DXA software.'

2) This could just be the review version, but some of the figures appear like they could use better clarity. However, it is understood that the pdf compilation versions of manuscripts often do not reflect the actual image. Just in case, be sure to run the figures through the PLoS figure/pace tool as recommended.

Reviewer #3: This study aimed to determine the cross-sectional associations of serum inflammation (CRP) with body composition and physical capacity in Canadian middle-aged and older adults with and without lower extremity OA, hand OA, and multimorbidities. The manuscript addresses an important and timely topic, particularly given the increasing prevalence of osteoarthritis and multimorbidity in aging populations. The focus on both body composition and physical capacity adds a valuable perspective to the existing literature, and the data analysis appears rigorous. I found the study to be highly interesting and relevant. However, I have a few suggestions for improvement and clarification, outlined in the attached file, that may help strengthen the clarity and impact of the manuscript.

Reviewer #4: Title and Abstract

Strengthen the title to emphasize the main finding

• Suggest:

"Associations of Body Fat Percentage with C Reactive Protein Levels in Older Canadians With and Without Osteoarthritis: Findings from the Canadian Longitudinal Study on Aging"

Clarify objective in the abstract

• Add more explicit statement about the strength of association and lack of OA specific effect in the conclusion.

Suggested Abstract Conclusion revision:

"Percent adiposity but not osteoarthritis status was consistently associated with systemic inflammation, suggesting that adiposity driven inflammation may contribute to OA related health outcomes."

Introduction

Clarify hypothesis formulation

• Currently, the hypotheses about interactions with OA status and sex are a bit broad. Clarify that the study is exploratory regarding interaction terms but was powered to test main effects.

Strengthen rationale for including multimorbidity

• Briefly explain why multimorbidity is important in the OA inflammation link. E.g., "Given the frequent co-occurrence of chronic conditions in aging, understanding whether inflammation is uniquely driven by OA or shared with other comorbidities is critical."

Methods

Participant Selection

• Provide flow chart or explicit mention of numbers excluded at each step (CONSORT-style).

Statistical Analysis

• Explicitly mention how missing data were handled.

o If excluded case-wise: "Analyses were conducted using complete case analysis; no imputation was performed."

Effect size criteria

• Clarify that "adjusted R² increases ≥1%" were used as thresholds for practical significance, in addition to p-values corrected for multiple comparisons.

Interaction terms

• Explain why OA subgroup interaction terms did not add value to model interpretation.

Results

Table 1

• Move all results tables (Table 1, S5) into a main results file or Supplementary File with clear captions.

• Summarize the main effects and emphasize effect sizes in the text.

Figures

• Ensure Figure 2 has clear axis labels, legend, and a description in the Results.

• Possibly combine or simplify figures to enhance clarity.

Sex differences

• Highlight key differences between females and males consistently across sections.

Strengthen Discussion of Clinical Relevance

• Relate the finding of small effect sizes (β ~0.13–0.16) to potential clinical impact on OA progression and management.

• Discuss limitations of CRP as a single inflammatory marker and suggest future studies exploring other biomarkers like IL-6, TNF-α.

Multimorbidity discussion

• Expand why the interaction term with multimorbidity did not yield significant changes in variance explained.

Limitations

• Include an explicit note on generalizability given the demographic profile (Canadian older adults) and the self-reported nature of OA diagnosis.

Financial Disclosure

Clarify that funders had no role

• Suggested sentence:

Add URLs of funders

• E.g.:

o CIHR: https://cihr-irsc.gc.ca

o Arthritis Society: https://arthritis.ca

Data Availability

State de-identified data are available

• Suggested statement:

"Data are available from the Canadian Longitudinal Study on Aging (www.clsa-elcv.ca) for researchers who meet the criteria for access to de-identified CLSA data."

Technical Edits

Check line numbers

• Some lines exceed 2-3 lines per sentence, consider splitting for readability.

Abbreviations

• Define all abbreviations on first use (e.g., OA, CRP, DXA).

Reference formatting

• Ensure references follow PLOS ONE’s citation style consistently.

**Do you want your identity to be public for this peer review?** For information about this choice, including consent withdrawal, please see our Privacy Policy

Reviewer #2: No

Reviewer #3: No

Reviewer #4: No

---

## [Author Response · Author response to Decision Letter 2]

25 Aug 2025

Please see the Response to Reviewers document for details addressing specific reviewer and editor comments.

---

## [Decision Letter · Decision Letter 2]

11 Jan 2026

Associations of body fat percentage with C-reactive protein levels in Canadian adults with and without osteoarthritis: Findings from the Canadian Longitudinal Study on Aging (CLSA)

PONE-D-24-17712R2

Dear Dr. Monica R. Maly,

We’re pleased to inform you that your manuscript has been judged scientifically suitable for publication and will be formally accepted for publication once it meets all outstanding technical requirements.

Kind regards,

Francesco Curcio, M.D., Ph.D.

Academic Editor

PLOS One

Additional Editor Comments (optional):

The manuscript is improved and it is suitable for publication

Reviewers' comments:

Reviewer's Responses to Questions

**Comments to the Author**

Reviewer #2: All comments have been addressed

Reviewer #5: All comments have been addressed

2. Is the manuscript technically sound, and do the data support the conclusions?

Reviewer #2: Yes

Reviewer #5: Yes

3. Has the statistical analysis been performed appropriately and rigorously?

Reviewer #2: Yes

Reviewer #5: Yes

4. Have the authors made all data underlying the findings in their manuscript fully available?

Reviewer #2: Yes

Reviewer #5: Yes

5. Is the manuscript presented in an intelligible fashion and written in standard English?

Reviewer #2: Yes

Reviewer #5: Yes

Reviewer #2: The reviewer recommendations have been adequately addressed and the paper now reads in a more clear and concise fashion. The paper adds to the literature and provides interesting insight into associations between adiposity and inflammation.

Reviewer #5: The Authors addressed all points of criticisms and I found manuscript greatly improved and suitable for publication in the present form

**Do you want your identity to be public for this peer review?** For information about this choice, including consent withdrawal, please see our Privacy Policy

Reviewer #2: No

Reviewer #5: No

---

## [Editor Report · Acceptance letter]

PONE-D-24-17712R2

PLOS One

Dear Dr. Maly,

I'm pleased to inform you that your manuscript has been deemed suitable for publication in PLOS One. Congratulations! Your manuscript is now being handed over to our production team.

Kind regards,

on behalf of

Dr. Francesco Curcio

Academic Editor

PLOS One